# Statins in Healthy Adults: A Meta-Analysis

**DOI:** 10.3390/medicina57060585

**Published:** 2021-06-07

**Authors:** Lee Sandwith, Patrice Forget

**Affiliations:** Institute of Applied Health Sciences, School of Medicine, Medical Sciences and Nutrition, Epidemiology Group, University of Aberdeen, Aberdeen AB25 2ZD, UK; leesandwith@ingfit.com

**Keywords:** statins, cardiovascular disease and mortality, cholesterol, primary prevention

## Abstract

*Background and Objectives*: In this paper, we investigated the efficacy of statin therapy on cardiovascular disease (CVD) reduction in adults with no known underlying health conditions by undertaking a meta-analysis and systematic review of the current evidence. *Materials and Methods*: We performed a systematic search to identify Primary Prevention Randomized Controlled Trials (RCTs) that compared statins with a control group where CVD events or mortality were the primary end point. Identified RCTs were evaluated and classified into categories depending on relevance in order to determine which type of meta-analysis would be feasible. *Results*: No differences were observed between categories with the exception of relative risk for all CVD events combined which showed a 12% statistically significant difference favouring studies which were known to include participants without underlying health conditions. Strong negative correlations between number-need-to-treat (NNT) and LDL-C reduction were observed for all Coronary Heart Disease (CHD) outcomes combined and all CVD outcomes combined. *Conclusions*: This project highlights the need for further research on the effects of statins on participants who do not suffer from underlying health conditions, given that no such studies have been conducted.

## 1. Introduction

People may be healthy while presenting hypercholesterolemia, and more specifically, increased Low Density Lipoprotein Cholesterol (LDL-C) levels. We do not know what strategy to adopt to prevent cardiovascular disease (CVD) in these cases. The critical issue for such people is that the standard of care for the management of high LDL-C is statin treatment; a practice which has become increasingly controversial due to side effects [1], claims that they have failed to substantially reduce CVD outcomes [2], statistical misrepresentation of the benefits [2] and financial conflicts of interest as a source of bias [2].

In terms of the evidence, there are hundreds of published randomised-controlled trials (RCTs) and several meta-analyses favouring statins [3,4,5,6]. Despite this, there is scepticism within the scientific community on the efficacy of statins, backed by a large number of scientific studies and publications [7,8,9,10,11]. The question of whether cholesterol plays a causal role in CVD is essential for proposing optimal strategies, not only pharmaceutical, but also dietetic.

The primary focus of the present work will be an investigation into the evidence on the benefits of statins on “otherwise healthy” participants. The justification for this is that the current standard of care for the management of high LDL-C is the ubiquitous prescription of statins, including participants with no underlying health conditions and where high LDL-C is the only exhibited CVD risk marker.

The key outcome of the study will be a review of the evidence on the efficacy of statins in reducing CVD risk in people with no underlying health conditions. We hypothesise that the CVD risk reduction effect size of statins is lower in participants without underlying health conditions compared to those with underlying health conditions. 

## 2. Methods

### 2.1. Search Protocol

MEDLINE, PubMed (1946–January 2020) and the Cochrane Database of Systematic Reviews were searched. Relevant studies were identified by using a combination of MeSH terms, text words and search equations focussing on the following terms: statins, cardiovascular disease and mortality, cholesterol, primary prevention and randomised controlled trials. All RCTs, reviews and references were examined to identify studies potentially eligible for inclusion.

### 2.2. Study Selection

We shortlisted studies for further investigation if they met the following criteria:Primary prevention;Placebo-based statins RCT;Mean follow-up of at least one year;Reported all-cause mortality (ACM), Coronary Heart Disease (CHD) or CVD events as primary, secondary or tertiary outcomes;Reported on known CVD risk confounding factors, such as LDL-C, age, sex, T2DM status, hypertension status and smoking status.

We designed a structured data abstraction form which included a comprehensive list of data points, and each study was categorised according to relevance (see Table 1). Titles and abstracts of all potentially eligible studies were then read in order to classify studies and where required papers were read in full. We also extracted information and assessed study quality using the Cochrane risk-of-bias tool.

### 2.3. Study Categorisation

All studies were evaluated to determine the type of outcome and population under investigation. Secondary prevention studies were not evaluated and categorised as they were not relevant to the research question. Primary prevention studies were classified into three categories as described in Table 1 with the objective of determining which type of meta-analysis would be feasible. 

### 2.4. Meta-Analysis

No Category 1 studies were found, meaning a standard aggregate meta-analysis was not feasible. Several Category 2 studies were found; however, although all studies reported sub-group data, none were sufficiently reported to enable a meta-analysis on the sub-group relevant to the present study. For example, studies may report sub-group data for “no hypertension”, however, within those sub-group data, participants may be included with other underlying health conditions, such as T2DM. This finding meant that a standard aggregate meta-analysis was not feasible.

### 2.5. Secondary Statistical Analysis and Meta-Regression Analysis

Given that meta-analyses were not possible, we decided to undertake a statistical analysis on two further levels. First, we compared the pooled results from the Category 2 and 3 studies to determine whether there was any difference between the two groups for the three outcomes below. Second, we conducted a meta-regression analysis to examine the relationships between number-needed-to-treat (NNT) and various confounding factors.

### 2.6. Outcomes

We performed a statistical analysis on the following three outcomes: Primary outcome: all-cause mortality; Secondary outcome: all CHD events combined;Tertiary outcome: all CVD events combined. 

Some aggregation was required to determine the secondary and tertiary outcomes:**Secondary outcome: All CHD events combined**
−Non-fatal or fatal MI−Death from coronary causes−Cardiac sudden death−Resuscitated cardiac arrest−Heart failure−Coronary angiography−Coronary artery bypass graft−Peripheral arterial surgery/angioplasty−Transient ischaemic attack−Percutaneous transluminal coronary angioplasty (PTCA) or Coronary artery bypass grafting/Percutaneous Coronary Intervention (CABG/PCI)−Revascularizations −Angina: unstable, stable or angina with evidence of ischemia−Interventional procedure**Tertiary outcome: All coronary and cardiovascular events combined**
−All secondary outcome events + −Fatal or nor fatal stroke−Death from cardiovascular causes

## 3. Results

### 3.1. Search Results

The search protocol identified a total of 143 relevant studies, with 97 RCTs and 46 reviews found. From the RCTs identified, 74 were disregarded as ineligible following a review of titles and abstracts, and 12 were duplicates. As such, 11 RCTs were deemed eligible for further, more comprehensive evaluation. A further 202 studies were found through a manual read-through of study references, reviews and meta-analyses. See Table 2, Table 3, Table 4 and Table 5 for further details. Figure 1 shows the logical flow used to classify RCTs identified through the search protocol.

### 3.2. Study Characteristics

Table 6 lists the RCTs included in the meta-analysis and further statistical analysis. This table includes relevant information pertaining to each study, including the type of statins studied, year of publication, mean follow-up and the number of participants included. In addition, this table includes data on potential confounding variables, such as average age, BMI, LDL-C at baseline, % LDL-C change, and smoking status as the percentage of participants with underlying health conditions, namely, hypertension and T2DM.

### 3.3. Meta-Analyses Comparing Category 2 and 3 Studies

Figure 2, Figure 3, Figure 4, Figure 5, Figure 6 and Figure 7 show relative risk ratios and absolute risk differences across Category 2 and 3 studies for the primary, secondary and tertiary outcomes. Stain therapy favoured control in both categories of study in terms of relative risk and risk difference for all outcomes (See Table 7 for effect sizes and statistical significance). No differences were observed between Category 2 and 3 studies with the exception of relative risk for all CVD events combined, which showed a 12% statistically significant difference favouring Category 2 studies, *X*^2^ (1, N = 88,877) = 4.06, *p* < 0.05 (Table 7, Figure 6).

### 3.4. Meta-Regression Analysis

Analysis was undertaken to examine the relationship between NNT and various confounding factors (Table 8). In order to achieve this, Spearman’s rank-order correlation was used, the results of which can be seen in Table 8 and in Figure 8, Figure 9, Figure 10 and Figure 11.

### 3.5. Primary Outcome

In terms of the primary outcome of all-cause mortality, no statistically significant correlations were observed between NNT and any LDL-C metrics (Figure 8, Figure 9, Figure 10 and Figure 11). There was a strong positive correlation between the percentage of smokers and NNT per year with regard to all-cause mortality (*r_s_*(13) = 0.703, *p* < 0.01) [Table 8]. 

### 3.6. Secondary Outcome

There was a strong negative correlation between the NNT per year for the secondary outcome of all CHD events combined and mean LDL-C reduction (mg/dL: *r_s_*(13) = 0.764, *p* < 0.01; mmol/L: *r_s_*(13) = 0.757, *p* < 0.01) [Figure 9 and Figure 10, respectively].

### 3.7. Tertiary Outcome

There was a strong negative correlation between the NNT tertiary outcome per year of all CVD events combined and the mean LDL-C reduction (mg/dL: *r_s_*(13) = 0.720, *p* < 0.01; mmol/L: *r_s_*(13) = 0.726, *p* < 0.01) and mean LDL-C % reduction (*r_s_*(13) = 0.699, *p* < 0.01) [Figure 9, Figure 10 and Figure 11, respectively].

## 4. Discussion

Through a meta-analysis of primary prevention studies, this study is not able to confirm any difference in statins efficacy between RCTs which included participants with and without underlying health conditions. This was with the exception of a 12% relative risk difference in the tertiary outcome of all CVD events combined favouring studies with the highest number of “otherwise healthy” participants. Should our hypothesis be true, we may have expected to see reduced efficacy in Category 2 studies, given that they did not primarily focus on participants with underlying health conditions. However, it is extremely difficult to draw conclusions from this result given that the percentage of “otherwise healthy” participants in each study is unknown and all studies included a significant percentage of participants exhibiting known heart disease risk factors. For example, 44% of participants in the WOSCOPS trial were smokers, and only 22% were reported as never having been smokers [12]. Similarly, 38% of HOPE-3 participants were reported as being hypertensive [15], and 21% of participants in MEGA were reported as having T2DM [14]. 

Meta-regression analysis showed an absence of an association between LDL-C reduction and NNT for the primary outcome of all-cause mortality. However, statistically significant negative associations between LDL-C reduction and NNT in the secondary and tertiary outcomes support a number of other meta-analyses that suggest that lowering LDL-C has a positive effect on reducing the risk of CHD and CVD events [3,4,5,25,26,27]. Correlation, however, does not prove causation, so such associations should be interpreted with significant caution. This is particularly pertinent to the results of statins trials, given the complexity associated with heart disease and the constantly evolving science. Central to this evolution is the emerging theory that LDL-C in its own right may not be as significant in the heart disease story as previously thought. For example, a wide range of potential heart disease risk markers are now considered to be more important than LDL-C, such as inflammation [28,29,30,31], LDL particle size [32,33,34], Apolipoprotein B [35] and LDL particle number [36], Lipoprotein(a) [37], lipoprotein ratios [38] and the ratio between triglycerides and HDL [39]. Inflammation specifically needs a special mention, as it has been the cause of much controversy in statins trials. For example, JUPITER reported statins as deriving a 20% RRD in all-cause mortality and almost 50% RRD in heart attacks and strokes. However, although an association with CRP reduction was shown, an important inflammation marker, no association was observed for LDL-C [31]. 

Conversely, it cannot be neglected that both naturally randomized genetic studies and randomized intervention trials have consistently shown that lowering plasma LDL particle concentration should reduce the risk of CVD events. This suggests that, through different approaches, LDL reduction lowers CVD risk, even though the effect of LDL increase has not been formally tested [25,26,27].

Our research contributes to the evidence that the results from statins trials may have been misrepresented. Most trials report either relative risk or hazard ratios and conclude that statins have significantly positive effects on CVD outcomes. However, although relative risk is one of a number of appropriate ways to present results, it can result in an overestimation of effect size, given that the underlying absolute risk is concealed [40]. JUPITER, for instance, reported that “*rosuvastatin significantly reduced the incidence of major cardiovascular events*”. As a specific example, the hazard ratio reported for myocardial infarction (MI) was 0.46, which could be interpreted as rosuvastatin reducing heart attacks by 54% compared to the control. This was based on 31 and 68 MIs in the intervention and control groups, respectively, unquestionably showing more than double the number of MIs in the control group. 

However, in terms of absolute risk, the difference between groups was only 0.42%, given the large sample size of almost 9000 in each study arm. Thus, whilst the study authors are technically accurate in reporting a significant difference between groups, the hazard ratio is potentially misleading without the context of absolute risk. In real-world terms, only 0.35% and 0.76% of participants suffered myocardial infarctions in the rosuvastatin and control groups, respectively, which is a miniscule effect size compared to that presented through a hazard ratio. In another example, HOPE-3 reported “*a significant reduction in the risk of cardiovascular events with the use of rosuvastatin*”. This was on the back of a hazard ratio of 0.76 for the coprimary outcome of a composite of death from cardiovascular causes, nonfatal myocardial infarction, or nonfatal stroke. The absolute risk difference for the same endpoint was 1.10%, deriving an NNT of 91. This common method of reporting has been highlighted in many critiques of statins trials, and is one of the factors which compounds statin scepticism [2,41,42,43,44].

This type of selective reporting may, in fact, be at least partially attributed to bias. Although all studies were found to be at low risk of bias after assessment using the Cochrane risk-of-bias tool, it must be acknowledged that there may be a risk of bias given that the majority of studies were funded by the pharmaceutical companies manufacturing the drugs under examination [45,46].

In conclusion, the efficacy of statins in reducing CVD risk in people with no underlying health conditions cannot be directly quantified. Our study has highlighted a significant gap in the literature regarding the efficacy of statins therapy on healthy adults with no underlying health conditions. It appears that the majority of RCTs conducted to date have focussed on participants exhibiting multiple heart disease risk factors, such as smoking, hypertension, obesity and diabetes. Although the majority of these studies paved the way for the current guidelines on statins therapy, none seem to have adequately isolated LDL-C as a standalone risk factor. From the studies that we identified as potentially having included a significant percentage of participants without underlying health conditions, none adequately reported this cohort in sub-group analyses to enable the present study to execute the intended aggregate meta-analysis. Given the evolving research on the relationship between LDL-C and heart disease risk, a review of the current guidelines is highly recommended. Statins clearly demonstrate benefits in terms of reducing CVD risk, that is, when the cardiovascular risk is high as a whole, but whether this is due to LDL reduction, or whether the statins are acting on other heart disease risk factors is not sufficiently understood. 

## Figures and Tables

**Figure 1 medicina-57-00585-f001:**
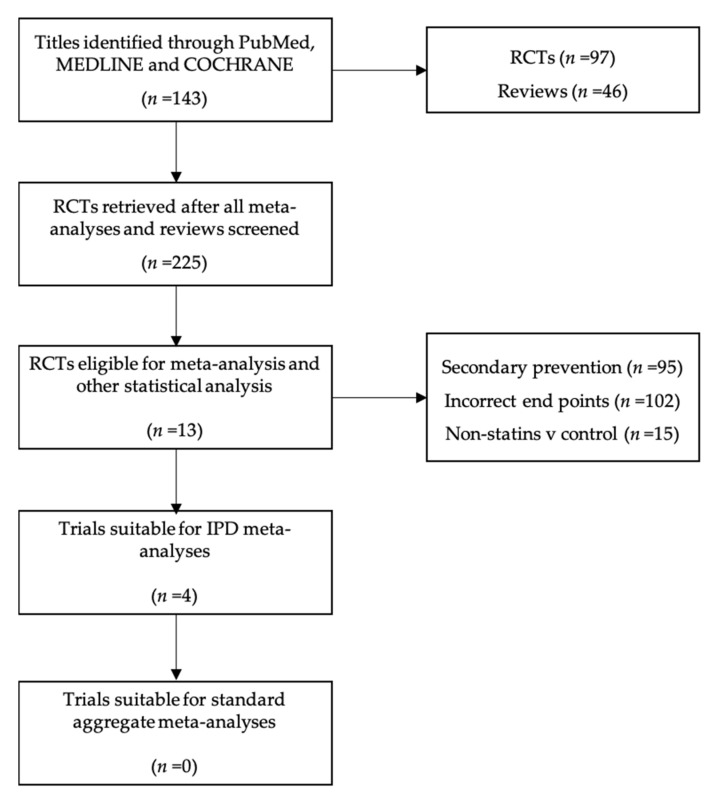
Trial workflow.

**Figure 2 medicina-57-00585-f002:**
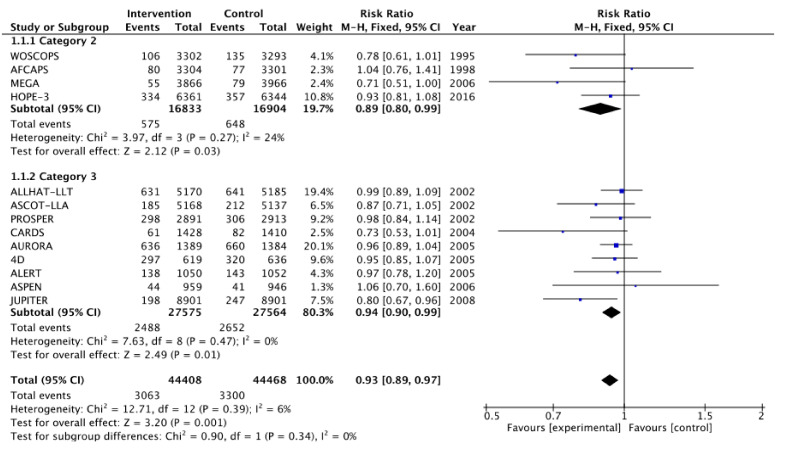
Relative risk across Category 2 and 3 studies for the primary outcome of all-cause mortality. Heterogeneity across studies was low, so fixed effects were used for the meta-analysis (Heterogeneity: Chi^2^ = 12.71, *p* = 0.39, I^2^ = 6%).

**Figure 3 medicina-57-00585-f003:**
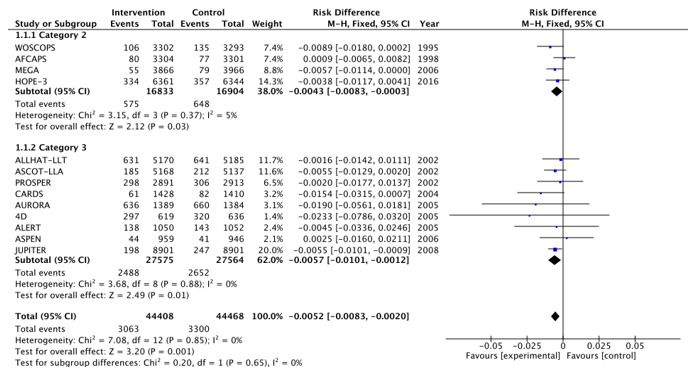
Risk difference across Category 2 and 3 studies for the primary outcome of all-cause mortality. Heterogeneity across studies was low, so fixed effects were used for the meta-analysis (Heterogeneity: Chi^2^ = 7.08, *p* = 0.85, I^2^ = 0%).

**Figure 4 medicina-57-00585-f004:**
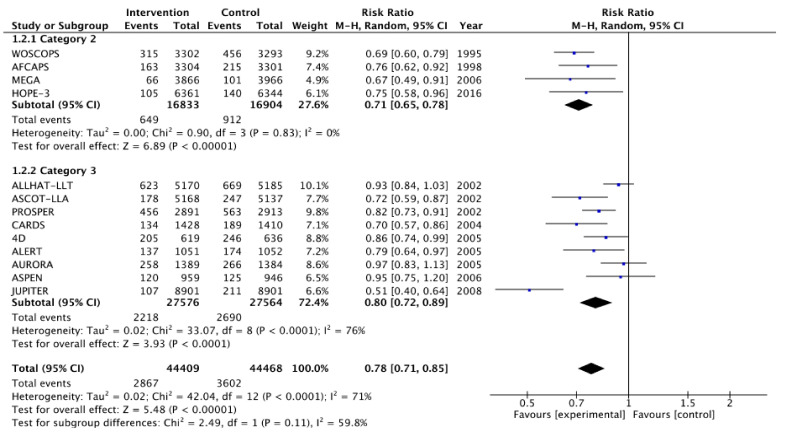
Relative risk across Category 2 and 3 studies for the secondary outcome of all CHD events combined. Heterogeneity across studies was high, so random effects were sed for the meta-analysis (Heterogeneity: Chi^2^ = 42.04, *p* < 0.001, I^2^ = 71%).

**Figure 5 medicina-57-00585-f005:**
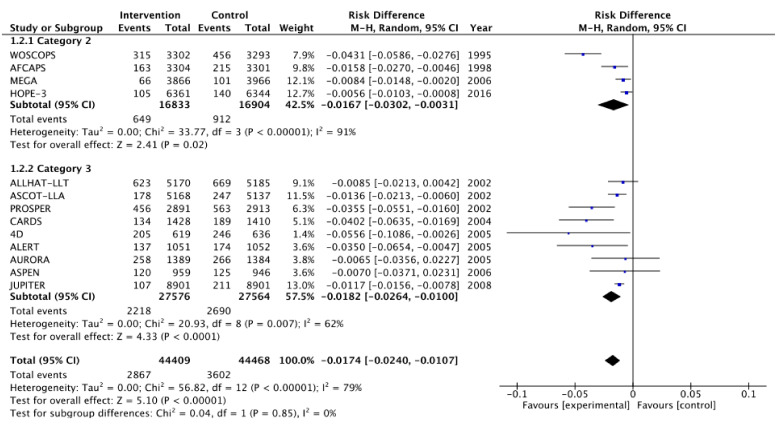
Risk difference across Category 2 and 3 studies for the secondary outcome of all CHD events combined. Heterogeneity across studies was high, so random effects were used for the meta-analysis (Heterogeneity: Chi^2^ = 56.82, *p* < 0.001, I^2^ = 79%).

**Figure 6 medicina-57-00585-f006:**
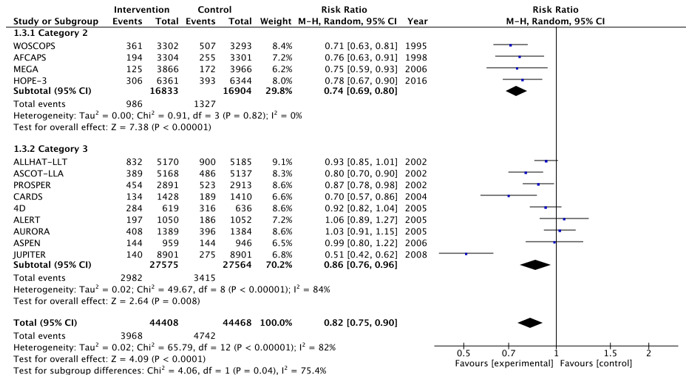
Relative risk across Category 2 and 3 studies for the tertiary outcome of all CVD events combined. Heterogeneity across studies was high, so random effects were used for the meta-analysis (Heterogeneity: Chi^2^ = 65.79, *p* < 0.001, I^2^ = 82%).

**Figure 7 medicina-57-00585-f007:**
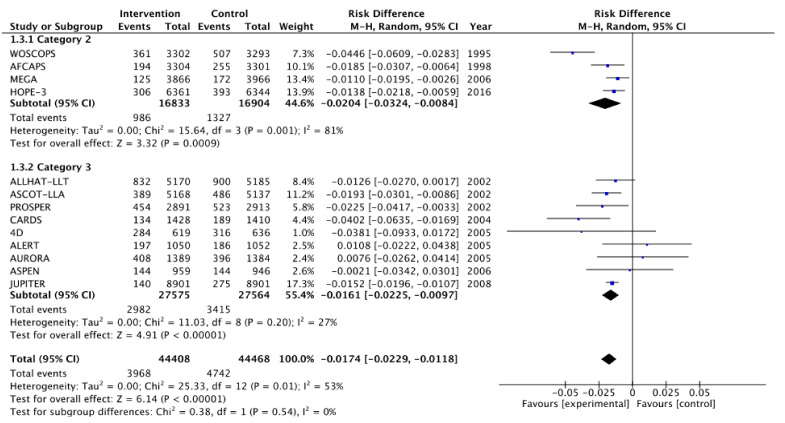
Risk difference across Category 2 and 3 studies for the tertiary outcome of all CVD events combined. Heterogeneity across studies was high, so random effects were used for the meta-analysis (Heterogeneity: Chi^2^ = 25.33, *p* < 0.05, I^2^ = 53%).

**Figure 8 medicina-57-00585-f008:**
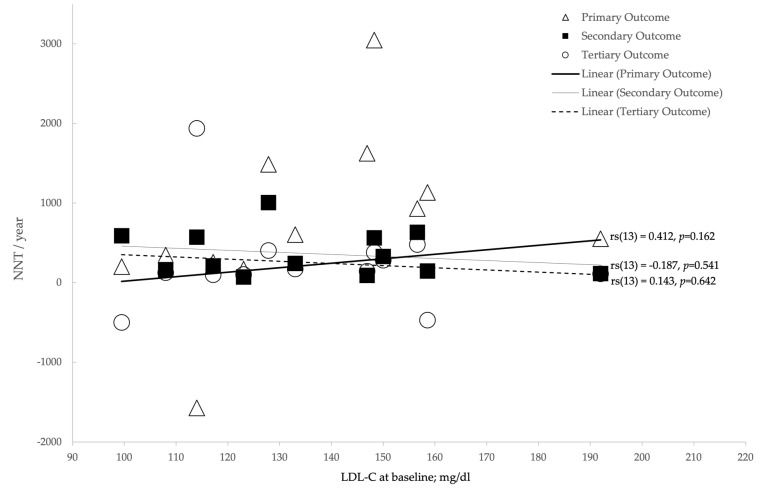
The relationship between LDL-C at baseline (mg/dL) and number-needed-to-treat (NNT)/year.

**Figure 9 medicina-57-00585-f009:**
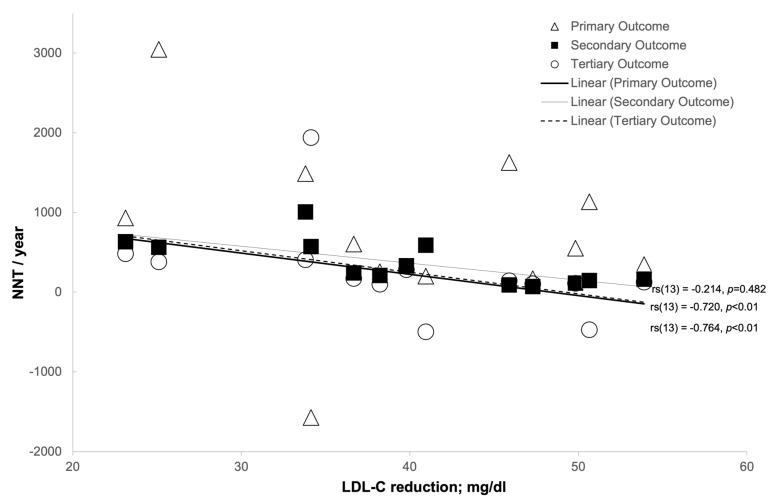
The relationship between LDL-C reduction (mg/dL) and number-needed-to-treat (NNT)/year.

**Figure 10 medicina-57-00585-f010:**
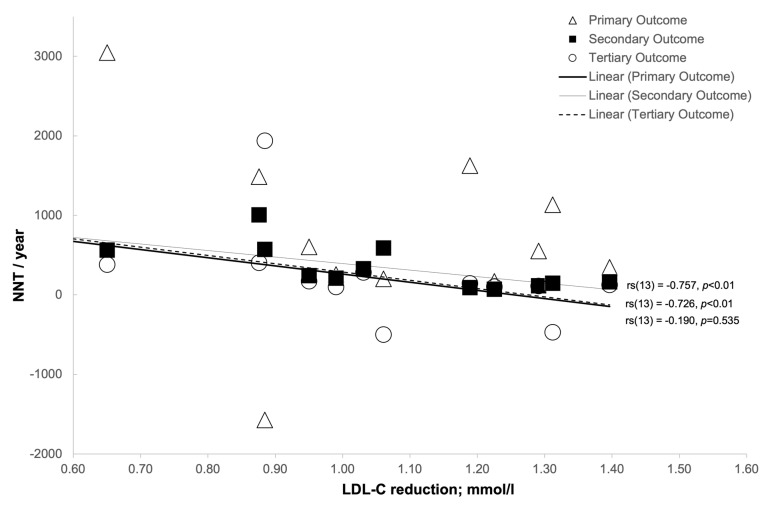
The relationship between LDL-C reduction (mmol/L) and number-needed-to-treat (NNT)/year.

**Figure 11 medicina-57-00585-f011:**
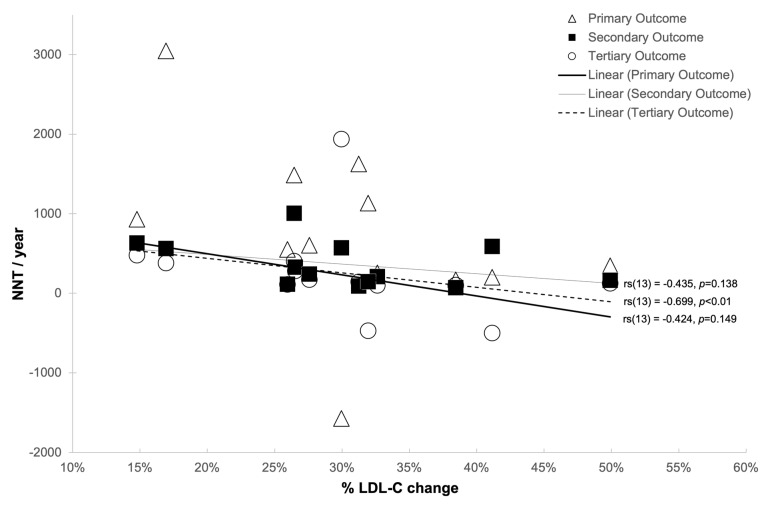
The relationship between LDL-C reduction % and number-needed-to-treat (NNT)/year.

**Table 1 medicina-57-00585-t001:** Randomised Controlled Trial (RCT) classification matrix.

Category	Description	Meta-Analysis Feasibility
Not relevant	Not an RCT, not statins versus control, secondary prevention	Not feasible
Category 1	Specifically studied participants with no underlying health conditions other than dyslipidaemia or hypercholesterolemia	Standard aggregate meta-analysis
Category 2	Included a significant percentage of participants without underlying health conditions	Sub-group meta-analysis
Category 3	Met all study eligibility criteria except that the specific focus was on participants with one or more underlying health conditions	Other statistical analyses

**Table 2 medicina-57-00585-t002:** Summary of search results.

Database	Study Design	Studies Retrieved	Category 2	Category 3
MEDLINE	Randomised Controlled Trial	22	3	3
PubMed	Randomised Controlled Trial	75	3	2
MEDLINE	Meta-analysis	30	-	-
PubMed	Meta-analysis	11	-	-
Cochrane	Meta-analysis	5	-	-
RCTs retrieved from search protocol for further evaluation		23		
Incremental RCTs retrieved from manual read of all reviews		202		
Total RCTs retrieved for further evaluation		225		

**Table 3 medicina-57-00585-t003:** Summary of studies retrieved split by primary prevention, secondary prevention and categorised by exclusion category and population. Category 1—100% without underlying health conditions or reported in sub-groups; Category 2—A percentage of participants included without underlying health conditions; Category 3—Underlying health conditions in 100% of participants. ^†^ May include some studies which were mixed primary and secondary; such studies were categorised as primary prevention. ^±^ CIMT, IMT, CAC, Calcium Volume Score, Atherosclerosis progression.

Category	n	%
Total number of RCTs retrieved	225	-
Primary prevention ^†^	130	58%
Secondary Prevention	95	42%
RCTs eligible for meta-analysis and other statistical analysis	13	10%
Category 1	0	0%
Category 2	4	31%
Category 3	9	69%
Primary studies by exclusion category	117	90%
Not statins v control	15	13%
Incorrect outcome	102	87%
Cholesterol levels	74	63%
Atherosclerosis markers ^±^	11	9%
Blood Pressure	4	3%
Other biomarkers	3	3%
CRP	2	2%
QUALY	1	1%
Bone mineral density	1	1%
Ischemic episodes	1	1%
Arterial inflammation	1	1%
Erectile function	1	1%
serum adiponectin levels	1	1%
Ventricular Diastolic Function	1	1%
Diabetic macular edema.	1	1%

**Table 4 medicina-57-00585-t004:** Category 1, 2 & 3 studies by health condition studied.

Health Condition Studied	n	%
None	4	31%
T2DM	3	23%
Hypertension	2	15%
Kidney disease	2	15%
Multiple comorbidities	1	8%
Inflammation	1	8%

**Table 5 medicina-57-00585-t005:** All primary studies by health condition studied.

Health Condition Studied	130	Pct
Hypercholesterolemia/Dyslipidemia	60	46%
T2DM	11	8%
Hypercholesterolemia/Dyslipidemia + another condition	7	5%
Multiple comorbidities	7	5%
Hypertension	6	5%
Hypertension and dyslipidemia	6	5%
None	4	3%
Diabetes	3	2%
High CHD risk	3	2%
Metabolic Syndrome	3	2%
kidney disease	2	2%
Stroke	2	2%
Acute Coronary Syndrome	1	1%
Arterial hypertension	1	1%
Atherosclerotic progression	1	1%
Bone mineral density	1	1%
CAPD	1	1%
Diabetic macular edema	1	1%
Dilated cardiomyopathy	1	1%
Familial Hypercholesterolemia	1	1%
HIV	1	1%
Inflammation	1	1%
Microalbuminuria	1	1%
Obesity	1	1%
Refractory Nephrotic Syndrome	1	1%
Stable angina pectoris.	1	1%
Statins users	1	1%
Subclinical hypothyroidism	1	1%

**Table 6 medicina-57-00585-t006:** Study characteristics.

	WOSCOPS	AFCAPS	MEGA	HOPE-3	4D	ASPEN	CARDS	ASCOT-LLA	ALLHAT-LLT	PROSPER	AURORA	ALERT	JUPITER
Year	1995	1998	2006	2016	2005	2006	2004	2002	2002	2002	2005	2005	2008
Statin	Pravastatin	Lovastatin	Pravastatin	Rosuvastatin	Atorvastatin	Atorvastatin	Atorvastatin	Atorvastatin	Pravastatin	Pravastatin	Rosuvastatin	Fluvastatin	Rosuvastatin
Number of participants	6595	6605	7852	12705	1255	1905	2838	10305	10355	5804	2773	1652	17802
Health condition(S) studied	None	None	None	None	T2DM	T2DM	T2DM	Hypertention	Hypertention	Multiple	Kidney disease	Kidney disease	Inflammation
Category of study	2	2	2	2	3	3	3	3	3	3	3	3	3
Men (%)	100%	85%	32%	54%	54%	62%	68%	81%	51%	48%	38%	66%	62%
Mean age at baseline (years)	55.2	58.0	58.3	65.8	65.7	60.5	61.6	63.1	66.3	75.3	64.2	48.5	66.0
Mean follow up (years)	4.9	5.2	5.3	5.6	3.93	4	3.95	3.3	4.8	3.2	3.8	6.7	1.9
Mean BMI at baseline	26.0	27.1	23.8	27.1	27.5	28.9	28.7	28.6	29.9	26.8	25.4	26.0	28.4
LDL status													
Baseline (mg/dl)	192	150	157	128	126	114	117	132	148	147	100	160	108
% reduction	26%	27%	15%	26%	38%	30%	33%	28%	17%	31%	41%	32%	50%
Reduction in mg/dl	50	40	23	34	47	34	38	37	25	46	41	51	54
Reduction in mmol/l	1.29	1.03	0.60	0.88	1.23	0.88	0.99	0.95	0.65	1.19	1.06	1.31	1.40
Smoking status													
Never smoked (%)	22%	NR	79%	72%	60%	NR	35%	NR	NR	NR	NR	NR	NR
Past smoked (%)	34%	NR	NR	NR	32%	NR	43%	NR	NR	NR	NR	NR	NR
Current smoker (%)	44%	12%	21%	28%	9%	13%	22%	33%	23%	27%	15%	17%	16%
Hypertention (%)	16%	22%	42%	38%	NR	52%	84%	100%	100%	62%	33%	75%	57%
Diabetes (%)	1%	4%	21%	6%	100%	100%	100%	25%	35%	11%	26%	17%	TBC

NR—Not reported.

**Table 7 medicina-57-00585-t007:** Relative Risk Reduction (RRD), Absolute Risk Difference (ARD) and Number Needed to Treat (NNT)/year in the trials for all of the present study’s outcomes. * Significant to *p* ≤ 0.05; ** significant to *p* ≥ 0.01.

	All-Cause Mortality			All CHD Events Combined			All CVD Events Combined		
Trial	% RRR	%ARD	NNT/year	% RRR	%ARD	NNT/year	% RRR	%ARD	NNT/year
WOSCOPS [12]	22% *	0.89%	551	31% *	4.31% **	114	29%	4.46% **	110
AFCAPS [13]	−4%	−0.09%	−5864	24%	1.58% **	329	24%	1.85% **	281
MEGA [14]	29%	0.57%	931	33%	0.84%	631	25%	1.10% **	480
HOPE-3 [15]	7%	0.38%	1487	25%	0.56%	1007	22%	1.38% **	405
4D [16]	5%	2.33%	168	7% **	0.85% **	71	7%	1.26%	103
ASPEN [17]	−3%	0.45%	−1574	28%	1.36% **	571	20%	1.93% **	1938
CARDS [18]	1%	0.16%	256	18%	3.55%	209	13%	2.25% *	98
ASCOT-LLA [19]	13%	0.55%	603	30%	1.89%	242	30%	4.02% **	171
ALLHAT-LLT [20]	−6%	−0.25%	3047	14%	5.56%	563	8%	3.81%	379
PROSPER [21]	4%	1.90%	1626	21%	3.50%	90	13%	−1.08%	142
AURORA [22]	27%	1.54%	200	3% *	0.65%	589	−3%	−0.76%	−499
ALERT [23]	20%	0.55%	1133	5%	0.70% **	146	1%	0.21%	−472
JUPITER [24]	2% *	0.20%	345	49%	1.17% **	163	49%	1.52% **	125
CAT 2 POOLED	11% *	0.43% **	233	29% **	1.67% **	60	26% **	2.04% **	49
CAT 3 POOLED	6% **	0.60% *	167	20% **	1.82% **	55	14% **	1.61% **	62

**Table 8 medicina-57-00585-t008:** Meta-regression analysis showing the relationships between NNT/year and confounding factors.

		Mean Age at Baseline	Mean BMI at Baseline	Mean LDL-C at Baseline	Mean LDL-C Change (mg/dL)	Mean LDL-C Change (mmol/L)	Mean LDL-C Change (%)	Pct Male	Pct Smokers	Pct with Hypertension	Pct with Diabetes
NNT Primary Outcome	Correlation Coefficient	0.379	−0.071	0.412088	−0.214	−0.190	−0.424	−0.353	0.703 **	0.452	−0.232
	Sig. (2-tailed)	0.201	0.817	0.162	0.482	0.535	0.149	0.237	0.007	0.140	0.467
	N	13	13	13	13	13	13	13	13	12	12
NNT Secondary Outcome	Correlation Coefficient	−0.022	−0.022	−0.186813	−0.764 **	−0.757 **	−0.435	−0.361	0.011	−0.210	−0.007
	Sig. (2-tailed)	0.943	0.943	0.541	0.002	0.003	0.138	0.226	0.972	0.512	0.983
	N	13	13	13	13	13	13	13	13	12	12
NNT Tertiary Outcome	Correlation Coefficient	0.110	0.297	0.142857	−0.720 **	−0.726 **	−0.699 **	−0.163	0.110	−0.014	−0.063
	Sig. (2-tailed)	0.721	0.325	0.642	0.006	0.005	0.008	0.596	0.721	0.966	0.845
	N	13	13	13	13	13	13	13	13	12	12

** Correlation is significant at the 0.01 level (2-tailed).

## Data Availability

All the data are included in the article.

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
