# Peer review of "Statins in Healthy Adults: A Meta-Analysis"

_medicina, 2021, doi:10.3390/medicina57060585_

Round 1

Reviewer 1 Report

This article entitled 'Statins in healthy adults. A meta-analysis' recommended a review of the current guidelines of statin to healthy adults by performing meta-analysis.

This research must contribute to enlighten for both previous studies regarding efficacy of statins in CVD and future RCT studies, but the reviewer recommends to revise minor things.

In Page 1 and 2
Full name of "CHD" was written differently in each pages
Please recheck and revise one of things.
(In page 1 : Coronary / In page 2 : Cardio)

In Page 21 Discussion Paragraph 1
The expression like "This study showed no difference in statins efficacy between RCTs which included participants without underlying health conditions" was improper as this study couldn't categorize numerous RCTs in Catergory 1. Above expression sounds like that previous studies could already be fufilled for the proof of the efficacy of statins properly. Please confirm and revise the expression for suggesting the intention more properly.

Reviewer 2 Report

The manuscript reviewed the evidence on the efficacy of statins in reducing CVD risk in people with no underlying health conditions and highlighted a significant gap in the literature regarding this topic. All research components are present and clearly stated. The experimental design is appropriate and procedures clear. The results are logically presented and states statistical significance of findings. Statement and conclusion are presented but need minor revision to correlate with data and link with goals

Reviewer 3 Report

In the conclusions, it should be mentioned that the indication of statins is carried out when the cardiovascular risk is high as a whole and an evaluation of it has been carried out with Score or other evaluations. They are not indicated, except for familial dyslipidemias with an isolated elevation of LDL-chol. Therefore, the indication of statins to supposedly healthy people should be very low if the recommendations are followed. It must be brought into the discussion.
On the other hand, the evidence for LDL-Col and the association with cardiovascular disease should be discussed more extensively.

Ference BA, Ginsberg HN, Graham I, Ray KK, Packard CJ, Bruckert E, et al. Low-density lipoproteins cause atherosclerotic
cardiovascular disease. 1. Evidence from genetic, epidemiologic, and clinical studies. A consensus statement fromthe European
Atherosclerosis Society Consensus Panel. Eur Heart J. 2017;38(32):2459–72. 
